# Overcoming power-efficiency tradeoff in a micro heat engine by engineered system-bath interactions

Sudeesh Krishnamurthy [1], Rajesh Ganapathy [2,3] & A. K. Sood[1,2] ✉

All real heat engines, be it conventional macro engines or colloidal and atomic micro engines, inevitably tradeoff efficiency in their pursuit to maximize power. This basic postulate of finite-time thermodynamics has been the bane of all engine design for over two centuries and all optimal protocols implemented hitherto could at best minimize only the loss in the efficiency. The absence of a protocol that allows engines to overcome this limitation has prompted theoretical studies to suggest universality of the postulate in both passive and active engines. Here, we experimentally overcome the power-efficiency tradeoff in a colloidal Stirling engine by selectively reducing relaxation times over only the isochoric processes using system bath interactions generated by electrophoretic noise. Our approach opens a window of cycle times where the tradeoff is reversed and enables the engine to surpass even their quasistatic efficiency. Our strategies finally cut loose engine design from fundamental restrictions and pave way for the development of more efficient and powerful engines and devices.

According to the second law of thermodynamics, reversible cyclic heat engines operating between two reservoirs at different temperatures attain the Carnot limit, $\eta_C$—the absolute maximum theoretical efficiency for a cyclic heat engine[1]. However, thermodynamic processes are reversible only when performed infinitesimally slowly, and hence the power delivered by such an engine is negligible. On the other hand, while finite engine cycle times allow extraction of useful work, irreversibility creeps into engine performance and decreases efficiency[2,3]. Thus, irrespective of the length scale over which they are designed to operate, all heat engines trade-off efficiency, $\eta$, in their quest to maximize power, $P$, and vice versa[4,5]. i.e., $P$ and $\eta$ cannot be simultaneously maximized and $P$ is maximal only for $\eta < \eta_q$, the quasi-static efficiency[6]. This fundamental postulate of finite-time thermodynamics has been the most basic challenge in engine design for over two centuries from thermal and nuclear reactors[7], petroleum fractionation[8], internal combustion engines, thermoelectric materials[9,10] to microscopic colloidal[11,12], and atomic[13] engines. While the performance in each of these individual scenarios are typically optimized based on empirical

tradeoff relations, theoretical studies have been able to derive a tradeoff relation only for systems where the engine bath interactions are Markovian[2,3]. Yet, power-efficiency tradeoff has been observed in all heat engines designed hitherto, irrespective of whether a functional form can even be derived, and strategies to overcome this remain unknown.

The indomitable challenge in overcoming such a trade-off lies in the mechanisms of heat transfer between the system and the reservoirs, which occurs through real thermal conductors with finite conductivity. Heat transfer through such conductors reaches the final value asymptotically over infinite time with a rate constant given by the inverse of the relaxation time, $\tau_R$. In the quasistatic limit where the cycle time, $\tau_{cycle} \gg \tau_R$, this is inconsequential, as heat supply matches demand, and the engine performs reversibly as expected. But at any finite $\tau_{cycle}$, demand exceeds supply, and irreversibility creeps into the engine operation, and it fails to perform the intended cycle. This basic limitation persists even in the fluctuation-dominated regime of colloidal engines[11,12,14,15], where $\tau_R$

[1]Department of Physics, Indian Institute of Science, Bangalore 560012, India. [2]International Centre for Materials Science, Jawaharlal Nehru Centre for Advanced Scientific Research, Jakkur, Bangalore 560064, India. [3]Sheikh Saqr Laboratory, Jawaharlal Nehru Centre for Advanced Scientific Research, Jakkur, Bangalore 560064, India. ✉e-mail: asood@iisc.ac.in

is reduced to the fundamental minimum time required for heat transfer using thermal fluctuations as dictated by the fluctuation-dissipation theorem. Unlike macro engines however, although such engines could now allow manipulating the reservoir using bacterial active noise[14], squeezed thermal reservoirs[16] and, engineered fluctuations[15], such strategies have also failed to overcome the $P-\eta$ tradeoff even if they can perform at $\eta > \eta_C$[15,17]. Nevertheless, over the last few years, theoretical proposals have explored extreme limits such as infinitely fast processes[18], exploiting a working substance near a critical point[19], infinite precision[20], diverging currents[21], and suggested that breaking the tradeoff might still be asymptotically possible. But realizing these limits in experiments is practically impossible. While reducing $\tau_R$ over the entire cycle might be an unrealizable demand, theoretical studies[22] on atomic systems have suggested that even decreasing it over select individual processes of a cycle by manipulating system-bath interactions might still allow the engine to overcome the $P-\eta$ tradeoff. Nevertheless, whether such interactions can be realized in experiments remains to be seen.

Here, we demonstrate that the $P-\eta$ tradeoff can be overcome in a colloidal Stirling engine at finite times by electrophoretically inducing system-bath interactions to reduce $\tau_R$ during the isochoric processes. Unlike their macroscopic counterparts explored hitherto, such colloidal engines are known to be capable of extracting heat from noise correlations in the reservoir[14,15,23], albeit under specific design considerations. We begin by recognizing the spatio-temporal scales of operation in which the effective noise experienced by the engine are correlated and non-Markovian. We demonstrate that the system-bath interactions are dependent on the Hamiltonian of the system under such conditions and allow us to engineer $\tau_R$ during isochoric processes. Driving the heat engine by utilizing the engineered $\tau_R$, we design and execute a protocol that overcomes the $P-\eta$ tradeoff. Finally, we trace the trajectory of system in the $k-T_{eff}$ plane at key $\tau_{cycle}$ and provide an intuitive explanation of the mechanism of overcoming the tradeoff.

## Results

### Non-Markovian characteristics of engineered noise
We first discuss the mechanisms by which our engine exploits noise correlations to generate system bath interactions that are used to tune $\tau_R$. Our engines were constructed with a charged colloidal microsphere in an optical trap set between the plates of a capacitor (Fig. 1a). Since the suspending medium is a polar solvent—de-ionized water, the charges on the microsphere are screened by a cloud of counterions. A potential difference, $V_{in}$ (such as in Fig. 1b) applied across the plates of the capacitor results in electrokinetic flows that drag the trapped colloidal particle. The relaxation times of such flows are highly dependent on the underlying mechanisms and can span a large range of timescales from $\approx 1\,\mu s$ to 10 ms[24–30] (see Supplementary Note 1 and Supplementary Fig. 1 on possible mechanisms). In Fig. 1b, we trace the average displacement of the trapped particle from its mean position on application of a constant $V_{in}$. The electrokinetic flows generated by the applied electric field displace the particle in a fixed direction and saturates over a timescale $\approx 5$–10 ms. We utilize this drag force in our experiments to apply engineered noise on the trapped colloidal particle.

Designing an engine protocol using the system in Fig. 1 requires definition of an effective temperature—possible only if the engineered noise used in our experiment is equivalent to fluctuations in an equilibrium system at timescales of operation. To mimic thermal fluctuations required to define such an effective temperature, we applied a Gaussian voltage noise to the plates of the capacitor. At timescales larger than the sampling time of such a noise (2 kHz noise and 500 Hz position sampling in Fig. 2a), the probability distribution of displacements of the colloidal particle along the electric field, $P(\Delta y)$ remains to be Gaussian (Fig. 2a). Further, we observed that the corner frequency of the power spectral density remained the same as that at zero field (Fig. 2b) at $2.22 \pm 4$ Hz and the particle dynamics was thus similar to a thermal reservoir at an elevated temperature (see Supplementary Note 1 and Supplementary Fig. 2). Physically, the energy transferred from the electrophoretic noise is essentially work. But, since the input energy fluctuations are similar to our engineered reservoir, we follow[12,15,31] and consider this as heat transferred from an effective reservoir. However, these fluctuations are temporally different from thermal noise which have velocity autocorrelations of the microsphere lasting only up to $\tau_b \approx 10\,\mu s$[32]. In Fig. 1b, the dotted line marks the time period (0.5 ms) after which $V_{in}$ is changed to a freshly sampled random value in our system and during this interval, unlike thermal noise, electrophoretic forces drive the particle ballistically along a set

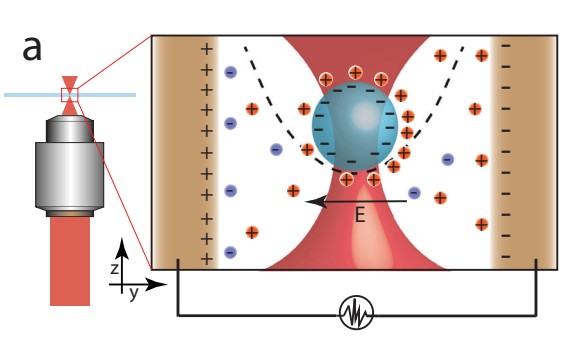

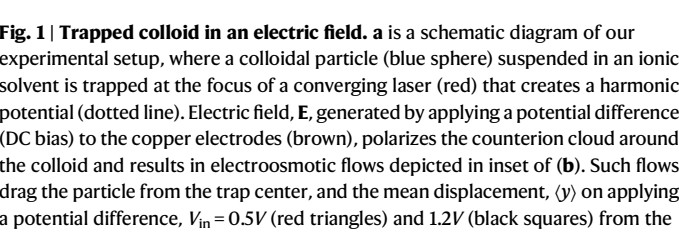

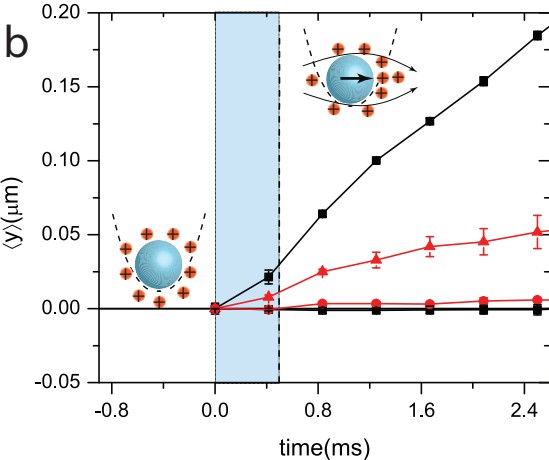

**Fig. 1 | Trapped colloid in an electric field. a** is a schematic diagram of our experimental setup, where a colloidal particle (blue sphere) suspended in an ionic solvent is trapped at the focus of a converging laser (red) that creates a harmonic potential (dotted line). Electric field, **E**, generated by applying a potential difference (DC bias) to the copper electrodes (brown), polarizes the counterion cloud around the colloid and results in electroosmotic flows depicted in inset of (**b**). Such flows drag the particle from the trap center, and the mean displacement, ⟨y⟩ on applying a potential difference, $V_{in} = 0.5V$ (red triangles) and 1.2$V$ (black squares) from the time of switching the field is plotted in (**b**). Also since the flows occur along **E**, displacements in the perpendicular direction ⟨x⟩ (red and black circles corresponding to $V_{in} = 0.5V$ and 1.2$V$) are within the limits of experimental error. The error bars correspond to standard error of mean over ≈7000 experiments. In our experiments, electrophoretic noise is generated by replacing $V_{in}$ by a freshly sampled random voltage every 0.5 ms marked by the dotted line. Within this time range (shaded blue), the particle is dragged in the direction set by the electric field and is the source of the non-Markovian behavior of electrophoretic noise.

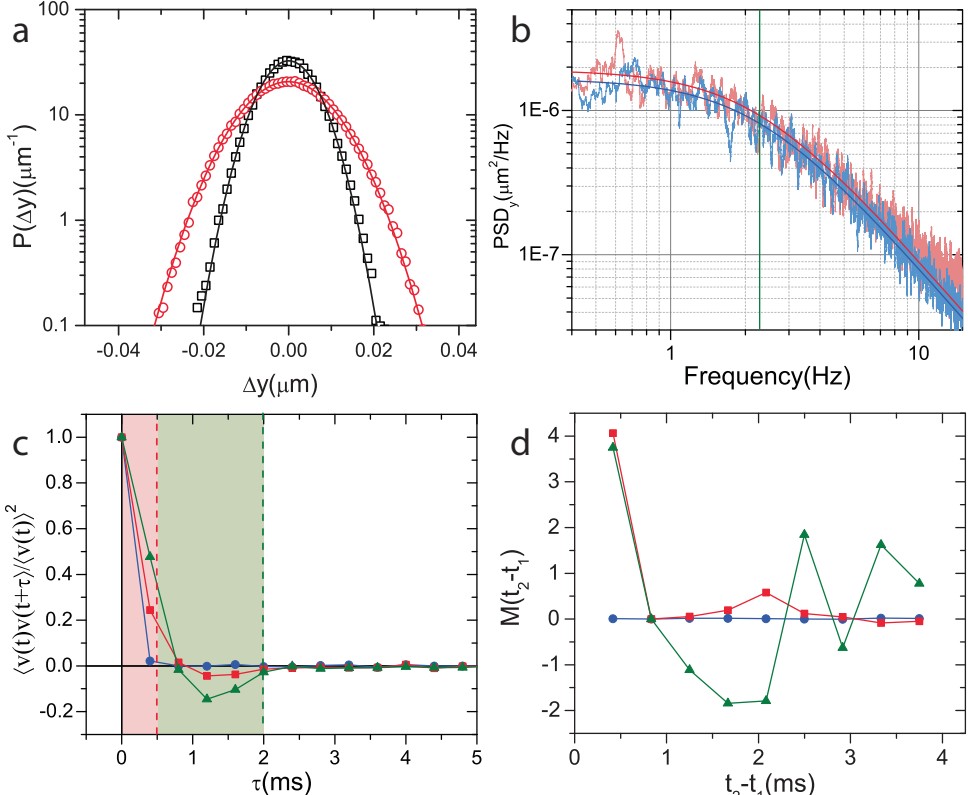

**Fig. 2 | Non-Markovian behavior of electrophoretic noise. a** is a plot of probability distribution of displacement of the particle, $P(\Delta y)$ with (red circles) and without (black squares) the applied voltage noise of $\langle|V_{in}|\rangle = 2.3V$. The distribution is calculated from $\approx$150,000 measurements of $P(\Delta y)$. The $P(\Delta y)$ were Gaussian (solid lines denote the Gaussian fits). **b** shows power spectral density of $y$ displacements before (blue line) and after (red line) applying the electric field. The solid lines represent Lorentzian fits for the data. **a, b** suggest that an effective temperature can be defined for our system. **c** shows velocity autocorrelation function, VACF before (blue circles) and after applying the 2 kHz (red squares) and 500 Hz (green triangles) voltage noise. While the force is delta correlated before applying the noise, it is correlated over the red shaded region for 2 kHz noise and red and green regions for the 500 Hz noise. **d** is plot of $M(t_2 - t_1)$ for constant $t_3$ and $t_2$ with $\Omega_1 = \Omega_2 = [-0.1, 0.15]$ before (blue circles) and after applying the 2 kHz (red squares) and 500 Hz (green triangles). $M(t_2 - t_1)$ is non-zero within the switching time of the applied noise. **c, d** suggest that the applied noise is non-Markovian. The analysis in (**b**–**d**) were performed on the data presented in (**a**).

direction. As a consequence, the velocity correlations of particle motion due to the engineered electrophoretic noise is non-zero during the sampling time 0.5 ms (Fig. 2c). The applied noise, thus, has a memory below the sampling time and is non-Markovian. To quantify the memory in the applied noise, we calculated the memory function, $M(t_2 - t_1)$[33] (see Supplementary Note 1 for detailed derivations). A non-zero value of the memory function implies the timescale at which the noise is non-Markovian. In our experiments, $M(t_2 - t_1)$ was observed to be high and non-zero in timescales less than the sampling time (Fig. 2d). Leveraging over this fundamental difference in timescales, we reduce $\tau_R$ during isochoric processes by utilizing these forces to nudge the system toward equilibrium.

**Engineering noise to reduce relaxation time**

While $\tau_s$ is a parameter characteristic to the input noise, $\tau_R$ also depends on dissipation and optical potential, which are specific to engine protocols. We first designed a Stirling engine and tuned these parameters to allow us to take advantage of the separation of timescales $\tau_b$ and $\tau_s$. A quintessential colloidal engine[11,12] utilizes the microsphere in Fig. 1a as a working substance and the harmonic optical potential as a piston. Synchronized variation of stiffness of the potential, $k$, and temperature, $T_{eff}$ as shown in Fig. 3a will then correspond to a mesoscopic equivalent to the conventional Stirling cycle[11]. In our experiment, these are controlled by input laser intensity and magnitude of the voltage noise applied to the capacitor, $\langle|V_{in}|\rangle$ respectively. Intuitively, performing the $k$ (volume) protocol at constant $T_{eff}$ is equivalent to an isotherm, and the change in $T_{eff}$ at

$k = k_{max}$ and $k_{min}$ (constant volume) to isochoric processes. Despite the separation of ballistic timescales $\tau_b$ and $\tau_s$, for $\tau_R$ to be different across these processes, engine-bath interactions should depend on the engine Hamiltonian[22] i.e., response of colloidal particle to electric field noise should depend on trap stiffness, $k$. We enumerated this response by the resultant effective temperature, $T_{eff} = k\langle y^2 \rangle/k_B$ where $k_B$ is the Boltzmann constant, caused by a fixed $\langle|V_{in}|\rangle$ at various $k$ (black squares in Fig. 3b). As $\tau_R = \gamma/k$ for thermal fluctuations[31,34] vanishes as $k \to \infty$, relaxations due to them outpace other mechanisms beyond a threshold $k$th. Engine-bath interactions are independent of engine Hamiltonian for such a noise and $T_{eff} = $ constant for $k > k$th. Such a saturation occurs in our system at $k_{th} \approx 4$ pN $\mu m^{-1}$ where $\gamma/k_{th} = 10$ ms $\approx \tau_s$. For $k < k_{th}$, however, electrophoretic forces start influencing the relaxations and $T_{eff} \sim \alpha \log k$. Intuitively, since the electrophoretic noise is freshly sampled every 0.5 ms $<< \tau_s = 10$ ms, $V_{in}$ is changed even before the particle explores the full extent of the system corresponding to the applied random voltage. As $k \to 0$, while the available extent to the system increases due to reduced confinement, the restriction on particle motion due to finite sampling remains the same and contribution of electrophoretic noise to $T_{eff}$ decreases. The exponent, $\alpha$ was tuned in our experiment by varying $\langle|V_{in}|\rangle$ (red squares in Fig. 3b) and $\tau_s$. Following Gouy-Chapman theory[27], $\tau_s$ and in turn $\alpha$ can be set using particle diameter (blue circles in Fig. 3b) and ion concentration (see Supplementary Note 2 and Supplementary Fig. 3). By reducing $\alpha$ using ion concentration, we can show that all results for thermal reservoirs can be retrieved (see Supplementary Note 2 and Supplementary Fig. 3). Our experiments

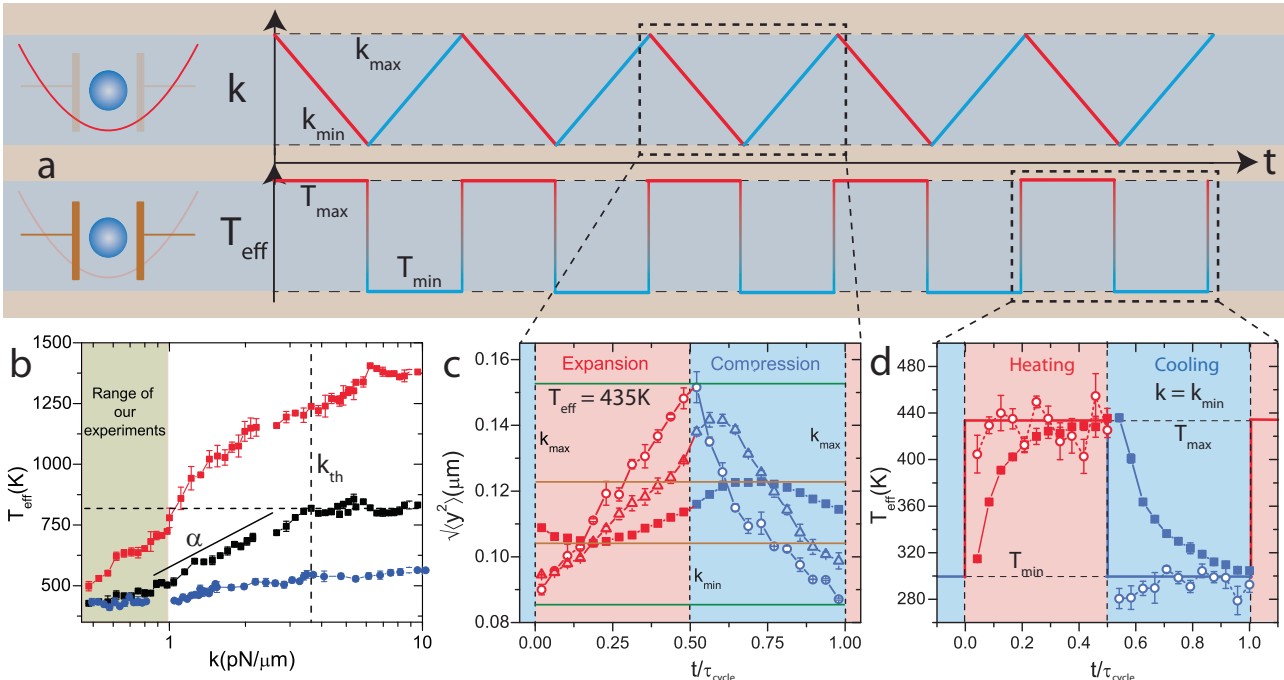

**Fig. 3 | Engineering relaxation times by utilizing system-bath interactions.**
**a** shows the time sequence of protocols in $k$ and $T_{eff}$ which correspond to executing a mesoscopic equivalent of Stirling cycle. **b** is a plot of $T_{eff}$ over 1.5 decades in $k$, where the $T_{eff}$ decreases as $k \to 0$ below a threshold $k_{th}$. The experiment for black and red squares were performed for microsphere of diameter 5 μm with different $\langle |V_{in}| \rangle$ such that $T_{eff}(k_{min}) = 427K$ (black squares) and 500K (red squares). Blue circles correspond to similar measurements for a 2 μm particle such that $T_{eff}(k_{min}) = 427K$. The exponent $\alpha$ at low $k$, which corresponds to the slope in the semilog plot and the threshold $k_{th}$ beyond which it saturates for the black squares are marked. Our engine was designed to operate in the green region where electrophoretic forces strongly influence the relaxations. **c** is a plot of $\sqrt{\langle y^2 \rangle}$, the equivalent of volume in our engine along rescaled time $t/\tau_{cycle}$ during the

isothermal processes, where only the $k$ protocol is performed at $T_{eff} = 435K$ for $\tau_{cycle} = 5$ s (open circles), 500 ms (triangles) and 100 ms (filled squares). On faster cycling, the volume explored by the particle drops from a maximum at $\tau_{cycle} = 5$ s (green line) to 25% of it at $\tau_{cycle} = 100$ ms (brown line) denoting a failure of equilibration. The relaxations in (**c**) are similar to those due to thermal noise plotted in Supplementary Fig. S8. **d** is a plot of $T_{eff}$ along $t/\tau_{cycle}$ along the isochoric processes, where only the $T_{eff}$ protocol is performed at $k = k_{min}$ for $\tau_{cycle} = 50$ s (open squares) and 50 ms (closed squares). The particle equilibrates to the final value in $\tau_R = 12$ ms for heating and $\tau_R = 17$ ms for cooling (as seen from data for 50 ms) and is significantly lower than 117 ms due to thermal noise. The averaging methods and estimation of error are discussed in Supplementary Note 4.

were performed in the shaded region in Fig. 3b marked by $(k_{max}, k_{min}) = (0.862 \pm 0.04, 0.322 \pm 0.07) \text{pN} \mu m^{-1}$ for $\langle |V_{in}| \rangle$ such that $T_{eff}(k_{min}) = 427K$, where the response of the system is influenced by electrophoretic forces.

To assess the effects of the electrophoretic forces on engine operation in the range of $k$ set in Fig. 3b, we determined $\tau_R$ along individual processes of the Stirling cycle. Along the isothermal processes, only the protocol in $k$ (Fig. 3a) is to be executed at constant $T_{eff}$. However, due to the dependence between $k$ and $T_{eff}$ noted in Fig. 3b, such a process cannot be performed with a constant $\langle |V_{in}| \rangle$. To execute the isothermal process, particularly the hot isotherm at $T_{eff} = 427K$, we modulated $\langle |V_{in}| \rangle$ to match the protocol in Fig. 3a (see Supplementary Note 3 and Supplementary Fig. 4). But, this modulation would then correspond to a series of relaxations and result in a larger $\tau_R$. In Fig. 3c, we plot $\sqrt{\langle y^2 \rangle}$, the equivalent of volume in our experiments along the rescaled time co-ordinate $t/\tau_{cycle}$ on performing the $k$ protocol at $T_{eff} = 435K$. Rescaling the time co-ordinate and averaging over multiple cycles gives a glimpse of $T_{eff}$ on continuous operation of the engine and allows us to represent experiments over different $\tau_{cycle}$ in the same scale (see Supplementary Note 4 and Supplementary Figs. 5 and 6 for details). At large $\tau_{cycle} = 5$ s, the system is close to the quasistatic limit and explores ≈95% of the available volume (difference between the green lines in Fig. 3c). As $\tau_{cycle}$ is reduced to 500 ms, the particle fails to equilibrate at $k_{min}$ and falls short of the maximum $\sqrt{\langle y^2 \rangle}$ at the end of expansion. In the compression cycle, $\sqrt{\langle y^2 \rangle}$ fails to decrease even as $k$ increases, and the maximum in $\sqrt{\langle y^2 \rangle}$ shifts away from $k = k_{min}$. At small $\tau_{cycle} = 100$ ms, this occurs at both $k_{max}$ and $k_{min}$ and the particle

explores only about 25% of the volume (difference between the brown lines in Fig. 3c) at $\tau_{cycle} = 5$ s. This, however, is the same $\tau_R$ as thermal noise and the response observed for a similar protocol performed for $T = 300K$ in the absence of electrophoretic forces closely matched Fig. 3c (see Supplementary Note 5 and Supplementary Fig. 7). Following the intuition used in Fig. 3b, modulation of $\langle |V_{in}| \rangle$ would compensate for the loss in volume explored by the particle due to fast sampling by adjusting the drive voltage appropriately, and volume relaxation would occur similar to thermal noise. Thus, utilizing the electrophoretic noise did not affect $\tau_R$ during isothermal processes. To observe the relaxations during isochoric processes, $T_{eff}$ protocol was performed at $k = k_{max}$ in Fig. 3d. If such a protocol was otherwise executed using only thermal noise, relaxation to 90% of the final $T_{eff}$ occurs over a timescale $\tau_I = 2.3\gamma/k_{min} = 270ms$[34] during both heating and cooling. Under the action of electrophoretic noise, however, fluctuations are no longer limited by dissipation in the system, and we observed $\tau_R = 12$ ms for heating and $\tau_R = 17$ ms for cooling, over which 90% of the relaxation occurred. A similar protocol performed at $k = k_{max}$ yielded $\tau_R = 9$ ms during both isochoric heating and cooling (see Supplementary Note 5 and Supplementary Fig. 8), while it would have relaxed to 90% of $T_{eff}$ in $\tau_h = 2.3\gamma/k_{max} = 99ms$ with thermal noise. Effectively, the addition of electrophoretic forces resulted in actively dragging the system closer to the final equilibrium states and reduced $\tau_R$ for isochoric processes. For comparison, in all macroscopic engines, $\tau_R$ is larger during the isochoric than isothermal processes[35,36] and in mesoscopic thermal engines[11], they are equal. By constructing a Stirling cycle using electrophoretically engineered engine-bath

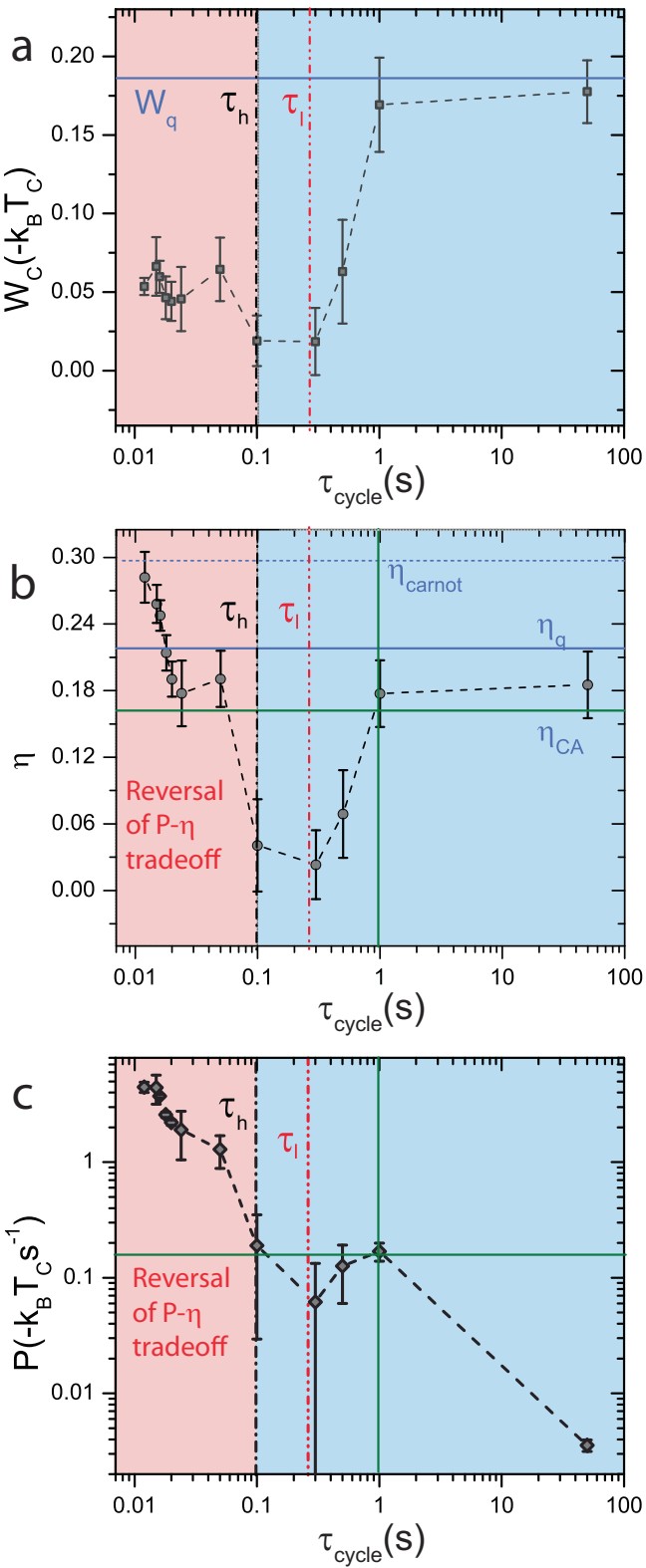

**Fig. 4 | Overcoming the power efficiency tradeoff at finite times.** Work done, $W_C$ (Squares), efficiency, $\eta$ (Circles) and power, $P$ (Diamonds) are shown for $\tau_{cycle}$ spanning over two decades above and below the relaxation times $\tau_h$ (black dash dotted line) and $\tau_l$ (red dash dot dot line) in (**a**–**c**) respectively. For $\tau_{cycle} > \tau_l$, $\eta$ decreases as $\tau_{cycle}$ is reduced while $P$ increases from its quasistatic value, as postulated by finite time thermodynamics and the region is marked blue. For $\tau_{cycle} < \tau_h$, both $\eta$ and $P$ increase simultaneously, thus overcoming the power efficiency tradeoff and the region is marked red. The data points in this region were averaged over an excess of 80,000 cycles. The averaging performed to obtain each data point and estimation of error are discussed in detail in Supplementary Note 4. The quasi-static limits $W_q$ and $\eta_q$ defined by equilibrium thermodynamics are represented as blue solid lines in (**a**, **b**). The short dotted line in (**b**) denotes the Carnot limit. The intersection of green lines in (**b**, **c**) represent efficiency at maximum power, $\eta^*$ and maximum in engine power for $\tau < \tau_h$.

microsphere obtained by tracking the particle through video microscopy. Work, $W$, and heat, $Q$ during the Stirling protocol are then calculated using the framework of stochastic thermodynamics[37] (see Supplementary Note 6 for detailed derivations). In the sign convention used in our experiment, work done by(on) the engine on(by) the reservoir is negative(positive) and heat transferred to(from) the engine from(to) the reservoir is negative(positive). Efficiency, $\eta = W_C/Q_H$, where $W_C$ is the mean work done per cycle and $Q_H$ is the average heat transferred from the hot reservoir. In Fig. 4, we plot $W_C$, $\eta$, and $P$ at various $\tau_{cycle}$, where the values are averaged over 30,000 cycles for $\tau_{cycle} < 100$ ms. At the large cycle time of $\tau_{cycle} = 50$ s, $W_C$ and $\eta$ are close to $W_q$ and $\eta_q$, predicted from equilibrium thermodynamics[11,14]. On decreasing $\tau_{cycle}$ however, as in the case of thermal engines, $W_C$ and $\eta$ decrease due to incomplete heat transfer. We anticipate deviations from this only as we approach, $\tau_{cycle} < \tau_l = 2.3\gamma/k_{min}$, above which 90% of equilibration occurs even with thermal noise at $k = k_{min}$. Strikingly, for $\tau_{cycle} < 100$ ms, $W_C$ increases and saturates at a fixed value (Fig. 4a). More importantly, $\eta$ continues to steadily rise and attain Carnot efficiency, $\eta_C$ within the limits of experimental error at $\tau_{cycle} = 12$ ms (Fig. 4b), the lowest $\tau_{cycle}$ used in our experiment (see Supplementary Note 4 for discussion on limitations). Also, $\langle\eta\rangle$ was 95% of $\eta_C$ at this $\tau_{cycle}$. Since $W_C$ saturates for $\tau_{cycle} < 100$ ms, $P$ continues to increase and has a finite value at $\tau_{cycle} = 12$ ms. Effectively, attaining higher power does not require us to trade-off efficiency in the range 12 ms $< \tau_{cycle} < 100$ ms. To compare this with existing theoretical formulations of finite time thermodynamics, we measured the efficiency at maximum power, $\eta^*$ for the region $\tau_{cycle} > \tau_l$. From Fig. 4c, maximum power is attained by the engine in this range at $\tau_{cycle} \approx 1$ s and the measured efficiency is $\eta^* = 0.17 \pm 0.03$ (Fig. 4b). The observed $\eta^*$ matches well with the prediction of Curzon-Ahlborn efficiency, $\eta_{CA} = 1 - \sqrt{T_C/T_H} = 0.162$[38] and with estimates of micro heat engine efficiency proposed by Schmeidl and Seifert $\eta^* = \frac{\eta_C}{2-\eta_C/2} = 0.161$[39] (see Supplementary Note 7 and Supplementary Fig. 9 for detailed discussions). It is to be noted that these theoretical formulations are derived for Markovian systems and do not capture the underlying phenomenon in our experiments. Yet, it is interesting that our results conform with the predictions in the region where $\tau_{cycle} > \tau_l$ below which we have engineered the relaxation times in our experiment.

## Origins of reversal of power-efficiency tradeoff
To elucidate the origins of the reversal of power-efficiency trade-off, we trace the path of the system in its state space, $k - T_{eff}$ plane for the Stirling cycle in Fig. 5. We divided the trajectory into 24 equal segments in each cycle and generated particle probability distributions, $P(\Delta y)$ by grouping corresponding $\Delta y$ over multiple cycles. Such $P(\Delta y)$ were observed to be Gaussian even at low $\tau_{cycle} = 15$ ms (see Supplementary Note 4 and Supplementary Fig. 6) and allowed us to define a $T_{eff}$ for all $\tau_{cycle}$. At large $\tau_{cycle}$, the state of the system should trace a rectangular path in the $k - T_{eff}$ plane for a Stirling cycle, and this was indeed

interactions and tuning the range of operation, we could engineer an inversion of relaxation times.

## Overcoming the power-efficiency tradeoff
After setting the experimental parameters and assessing their effects on $\tau_R$, we performed the Stirling cycle—which involves synchronously executing both the $k$ and the $T_{eff}$ protocols of Fig. 3a. The state of the system is described by the position and velocity of the trapped

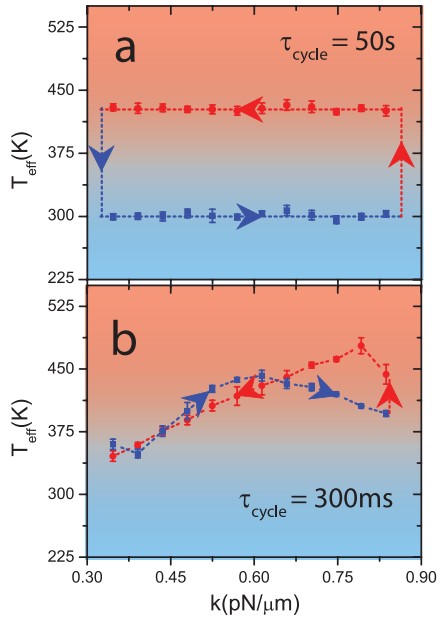
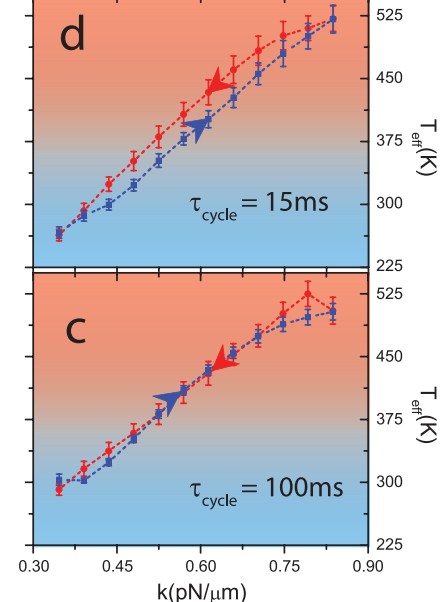

**Fig. 5 | Origins of the engineered reversal in power-efficiency tradeoff.**
**a**–**d** represent the state of the system on the $k − T_{eff}$ plane for $\tau_{cycle} = 50$ s, 300 ms, 100 ms, 15 ms respectively, where, the red circles are state points in contact with hot reservoir ($T_{max}$) and blue squares with cold reservoir ($T_{min}$). $\tau_{cycle} = 50$ s, plotted in (**a**), corresponds to the quasistatic Stirling cycle. $\tau_{cycle} = 300$ ms and $\tau_{cycle} = 100$ ms, plotted in (**b**, **c**), represent the state of the system at $\tau_l$ (just before the reversal) and $\tau_h$ (just after the reversal of the tradeoff). $\tau_{cycle} = 15$ ms, plotted in

(**d**), represents the system at very low $\tau_{cycle}$, where $\eta$ is close to $\eta_C$. The dotted lines are a guide to the eye with the arrows denoting the direction of progress of Stirling cycle. The trajectory for $\tau_{cycle} = 50$ s was averaged over 200 cycles, $\tau_{cycle} = 300$ ms over 27,000, $\tau_{cycle} = 100$ ms over 81,000, and $\tau_{cycle} = 15$ ms over 540,000 cycles respectively. The averaging methods and estimation of error are discussed in Supplementary Note 4.

observed at $\tau_{cycle} = 50$ s in our experiments (Fig. 5a). To investigate the cause for reversal in power-efficiency trade-off, we examined similar plots for $\tau_{cycle} = 300$ ms which is close to $\tau_l = 2.3\gamma/k_{min} = 270$ms (Fig. 5b) and $\tau_{cycle} = 100$ ms which is close to $\tau_h = 2.3\gamma/k_{max} = 99$ms (Fig. 5c). At $\tau_{cycle} = 300$ ms, the isotherms fail to equilibrate only at $k_{min}$. Quick compression(expansion) and failure of heat transfer from(to) the system is equivalent to adiabatic Joule heating(cooling) and $T_{eff}$ increases(decreases) at $k_{min}$ (Fig. 5b). Along the cold isotherm, as $k \rightarrow k_{max}$, the system recovers from such a failure, and $T_{eff}$ decreases. However, even at the end of the cold isotherm, $T_{eff}(k_{max}) > T_{min} = 300$K, and isochoric heating now leads to a higher $T_{eff}$ at the beginning of the hot isotherm. At $\tau_{cycle} = 100$ ms $\approx \tau_h$, the isotherms fail at both $k_{max}$ and $k_{min}$ (Fig. 5c). As volume explored by the system decreases significantly for $\tau_{cycle} < \tau_h$ (Fig. 5d), $\langle y^2 \rangle$ is almost constant and $T_{eff} \propto k$ along the isotherms (Fig. 5d). Nevertheless, the nature of system-bath interactions reduces $\tau_R$ along the isochores and ensures that the isothermal expansion (red line in Fig. 5d) has a higher $T_{eff}$ than the compression (blue line in Fig. 5d). However, this would not be the case if $\tau_R$ was higher due to additional ions in the solution (see Supplementary Note 2 and Supplementary Fig. 3). For engines where the isochores are otherwise performed using thermal baths, $\tau_R$ is even higher and $\approx \tau_h$, change in $T_{eff}$ across the isochores would then be negligible and $T_{eff}$ along isothermal compression is higher than expansion (see Supplementary Note 8 and Supplementary Fig. 10). For $\tau < 12$ ms (not accessible in our system), where even the isochores fail to equilibrate, this would also be the ultimate fate of our system. In fact, a critical examination of this fate of the thermal engine served as the motivation in engineering the right regime of operation for our engine (see Supplementary Note 8). Nonetheless, the reversal of the power-efficiency trade-off allowed us to attain $\eta_C$ at finite $P$.

An obvious corollary to the mechanism of overcoming the $P − \eta$ tradeoff is that the effect is limited to only a range of $T_{max}$ for a given $T_{min}$. A typical Stirling cycle driven by only two reservoirs is always less efficient than the Carnot cycle as the heat drawn from the hot bath

during the isochoric process is irreversible. In the case of the engine cycle in Fig. 5d, this heat is no longer required as fast compression during the cold isotherm would result in $T_{eff} > T_{max}$ and $\eta > \eta_q$ at this $\tau_{cycle}$. Intuitively, a total absence of heat transfer during cold isotherm noted in Fig. 5b–d would result in a constant $P(\Delta y)$ during the process. The corresponding maximum temperature at $k_{max}$ would be $(k_{max}/k_{min})T_{min} \approx 800$ K in our experiments. If $T_{max}$ were to be greater than this temperature, the engine cycle would require heat transfer during the isochoric process during low $\tau_{cycle}$ and would eventually be less efficient than $\eta_q$. The upturn in $\eta$ demonstrated in Fig. 4 would thus reduce at a higher $T_{max}$. We verified this corollary in our experiments by operating heat engines with a higher $T_{max} = 935$ K (see Supplementary Note 9 and Supplementary Figs. 11–14).

## Discussion

In conclusion, we have designed a colloidal Stirling engine that overcomes the power-efficiency tradeoff by utilizing electrophoretic noise to induce system-bath interactions that depend on the engine Hamiltonian. We tuned the range of engine operation to exploit such interactions and reduced $\tau_R$ during the isochoric processes, which resulted in the reversal of the $P − \eta$ trade-off. At the lowest $\tau_{cycle} = 12$ ms, our engine surpassed its quasistatic efficiency and even achieved $\eta_C$ at a finite $P$ within the limits of experimental error. The key requirement to our strategy to overcome such a basic limitation is the longer ballistic time of the driving noise, which could alternately be accomplished using optical[15,31], magnetic[40] and for that matter any chemical process that burns a fuel[41,42]. Despite this, reversal of the tradeoff can be achieved with such driving only by following the methods described above and setting the system parameters to enable assisted relaxation. Unlike our engineered reservoir, where additional work is required to maintain the state of the reservoir, such methods might not necessarily require a work input and might even be derived from any passive reservoir with a long correlation time such as viscoelastic baths[43]. In fact, recent theoretical studies on viscoelastic baths[43] have also

pointed to such reversals in tradeoff. Further, recent studies[44] have noted that a key feature of such active baths is that they contribute informational excess entropy to the engine, which in our case is zero (see Supplementary Note 6 for discussions). Thus, our effective reservoir might not transfer the input work required to maintain it to the heat engine. Since the system is driven out of equilibrium during the isochoric processes, our results do not contradict the universal trade-off relations derived under different conditions using Markovian dynamics[2,3]. Nevertheless, our results compare well with the observations of efficiency at maximum power[38,39] in the range of $\tau_{cycle} > \tau_h$ (see Supplementary Note 7 and Supplementary Fig. 9). But extending this similarity beyond this range to $\tau_{cycle} < \tau_h$ is not possible. As such, our experiments show that the $P - \eta$ tradeoff can be reversed for periodic heat engines by tailoring system-bath couplings. Reversing the $P - \eta$ tradeoff finally breaks free engine design from the fundamental restrictions postulated by finite-time thermodynamics. Our study now paves the way for extending this strategy to design better and more efficient thermoelectric devices[9,10] and NEMS heat engines[45].

## Methods

### Laser trapping and particle position determination

The micrometer-sized Stirling engine was realized using cross-linked poly(styrene/di-vinyl benzene) (P[S/DVB]) colloidal particles of 5 μm, which were obtained from Bangslabs, USA. The harmonic potential used to trap these particles was generated by focusing an IR laser beam of wavelength 1064 nm with a 100X Carl Zeiss objective (1.4 N.A.) mounted on a Carl Zeiss Axiovert microscope. The laser beam was generated by a Spectra Physics NdYVO4 laser head. The particles were suspended in deionized water at extremely low concentrations of a few particles per microliter. The trapped particle was imaged using a Basler Ace 180 kc color camera at a frame rate of 500 frame s$^{-1}$ for cycles with $\tau > 300$ ms. To generate sufficient statistics at low $\tau$, $\tau = 100$ ms, 50 ms, 24 ms were imaged at 1000 frame s$^{-1}$, $\tau = 20$ ms at 1200 frame s$^{-1}$, $\tau = 18$ ms at 1333 frame s$^{-1}$, $\tau = 16$ ms at 1500 frame s$^{-1}$, $\tau = 15$ ms at 1600 frame s$^{-1}$ and $\tau = 12$ ms at 2000 frame s$^{-1}$. The particle was tracked from the resulting images to sub-pixel resolution using custom made Matlab codes to an accuracy of 10 nm. $W_C$ and $\eta$ were calculated from the particle positions using the framework of stochastic thermodynamics[14,37].

### Temperature and volume control protocol

The $T_{eff}$ protocol was realized in our experiment by applying a Gaussian voltage noise sampled at 2 kHz generated using an Agilent 33521A waveform generator to custom made copper electrodes. The electrodes were made using copper wires of diameter 60 μm and were separated by a distance of ≈200 μm. During our experiment, we observed that the $T_{max}$ experienced by the particle was constant over span of 50 μm between the electrodes and the colloidal particle was trapped at the center of this region. Calculation of $\eta$ required that the changes in $k$ are performed slower than the timescale in which the particle experiences thermal noise and the additional electrical noise in our case. To this extent, we ensured that at the most 1/24th of the cycle was completed before a particle position was measured and a noise signal was generated afresh, less than previous realizations in literature[12] (see Supplementary Note 4).

The $k$ protocol in our experiment was realized by modulating the intensity of the trapping laser beam by an Intra-Action Corp acousto-optic modulator. The acousto-optic modulator was driven by an 40 MHz crystal oscillator, whose power in turn was modulated by a Ramp signal generated from a Tektronix AFG3021C waveform generator. The response time for changes in laser power was 200 ns.

## Data availability

All data are available in the manuscript or the Supplementary Materials.

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

## Acknowledgements
A.K.S. and S.K. thank the Department of Science and Technology, India for financial support under the Year of Science Fellowship.

## Author contributions
S.K.: conceptualization, investigation, methodology, formal analysis, data curation, software, validation, visualization, writing—original draft, writing—review and editing. R.G.: validation, writing—review and editing. A.K.S.: conceptualization, funding acquisition, methodology, project administration, resources, supervision, data curation, validation, writing—review and editing.

## Competing interests
The authors declare no competing interests.
