## [Peer Review File · Nature Communications]

nature portfolio

Peer Review File

TitleEditorial Note: This manuscript has been previously reviewed at another journal that is not operating a transparent peer review scheme. This document only contains reviewer comments and rebuttal letters for versions considered at *Nature Communications*.

Reviewers' Comments:

Reviewer #2:

Remarks to the Author:

I have reviewed this manuscript before. The Authors have replied to most comments raised by other Reviewers and myself. I am glad that the Authors have relaxed their claims, and now they better explain the main point of their work. Even if some points of the system remain unclear, I think this work is relevant for a broad community, and somehow will spark further work in this area.

Therefore, I am happy to recommend this paper for publication after the Authors fix the following issue. It is fundamental since it is related to the timescale of the applied noise, which the Authors incorrectly identify as $\tau_s = \lambda_D a/D$, λ_D being the Debye length, a the radius of the particle and D the diffusion coefficient of ions. This time scale does not apply in this situation and has nothing to do with the problem. It appears in induced charge electrokinetic phenomena, but these are effects that appear in the case of metallic particles and that are absent in the present case. A similar timescale that might play a role is the charge relaxation at the electrodes, since there is a buildup of charge upon application of voltage difference that partially screens the electric field in the solution (10.1103/PhysRevE.70.021506), but guess that the screening effect will be weak and quite slow in the case of de-ionized water, where there are not too many ions available.

In order to understand how the counterion cloud around the particle evolves after an electric field is applied, I guess it would be better to follow this reference:

<https://pubs.acs.org/doi/full/10.1021/la0607252>

Here, you can see that there are at least 3 effects (relaxations) that affect the steady-state value of the electrophoretic mobility: inertial relaxation at short times, then the Maxwell-Wagner (MW) relaxation, and the alpha relaxation. The alpha relaxation is the one that is related to the re-organization of ions around the particle, it is the slowest one (it scales as $\tau_{\alpha} \propto a^2/D$), but has little effect on the mobility. Then follows the MW, which scales as $\tau_{MW} \propto \lambda_D^2/D$, and finally the inertial one, which is related to the inertia of the fluid and happens at very short times. For the case of deionized water, the ions would be mainly H^+ and OH^- , whose diffusion coefficients are of the order of $5 \cdot 10^{-9} \text{ m}^2/\text{s}$, returning a $\tau_{\alpha} \approx 1 \text{ ms}$ for the $5 \mu\text{m}$ particles used in the experiment. The other timescales are shorter. I guess the problem is still more complicated, and the relaxation in the trap will be more involved. There are a few publications on the electrophoresis of trapped particles that could be considered for a better understanding of the problem:

<https://doi.org/10.1016/j.jcis.2009.05.017>

<https://doi.org/10.1063/1.2884147>

<https://doi.org/10.1002/elps.201300214>

<https://doi.org/10.1063/1.4882419>

<https://doi.org/10.1103/PhysRevE.90.032116>

I understand that this is too much information to be included in the manuscript, but I definitely ask the Authors to at least indicate that the problem is more complex than just the single timescale they wrongly identify.

Reviewer #3:

Remarks to the Author:

As indicated in my previous reports in Nature Physics, I find the results, methodology, and impact of the paper suitable for publication in Nature Communications, and even more after the new

modifications in the manuscript.

The authors have successfully tackled most of my previous queries and improve the clarity and interpretation of their excellent experimental results. Now it is more clear the theoretical background and why the results are interesting for a broad class of physicists.

The manuscript has gained a lot especially in what refers to quantification of non-Markovianity, which is key, and is now provided in terms of a recently-introduced measure.

I just have one suggestion for the authors, namely include Fig S2C in the Main Text, either as a panel or as an inset, or even the three panels in S2. I find the result in S2C excellent, and a perfect fingerprint of non-Markovianity in their experimental setup, which is a central assumption of the authors' claim, and in my opinion deserves to be revealed somewhere in the Main Text.

Once this modification is implemented I am ready to recommend the article for publication in Nature Communications.

REVIEWER COMMENTS

Reviewer #2 (Remarks to the Author):

I have reviewed this manuscript before. The Authors have replied to most comments raised by other Reviewers and myself. I am glad that the Authors have relaxed their claims, and now they better explain the main point of their work. Even if some points of the system remain unclear, I think this work is relevant for a broad community, and somehow will spark further work in this area.

Therefore, I am happy to recommend this paper for publication after the Authors fix the following issue. It is fundamental since it is related to the timescale of the applied noise, which the Authors incorrectly identify as $\tau_s = \lambda_D a/D$, λ_D being the Debye length, a the radius of the particle and D the diffusion coefficient of ions. This time scale does not apply in this situation and has nothing to do with the problem. It appears in induced charge electrokinetic phenomena, but these are effects that appear in the case of metallic particles and that are absent in the present case. A similar timescale that might play a role is the charge relaxation at the electrodes, since there is a buildup of charge upon application of voltage difference that partially screens the electric field in the solution (10.1103/PhysRevE.70.021506), but guess that the screening effect will be weak and quite slow in the case of de-ionized water, where there are not too many ions available.

In order to understand how the counterion cloud around the particle evolves after an electric field is applied, I guess it would be better to follow this reference : <https://pubs.acs.org/doi/full/10.1021/la0607252> Here, you can see that there are at least 3 effects (relaxations) that affect the steady-state value of the electrophoretic mobility: inertial relaxation at short times, then the Maxwell-Wagner (MW) relaxation, and the alpha relaxation. The alpha relaxation is the one that is related to the re-organization of ions around the particle, it is the slowest one (it scales as $\tau_\alpha \simeq a^2/D$), but has little effect on the mobility. Then follows the MW, which scales as $\tau_{MW} \simeq \lambda_D^2/D$, and finally the inertial one, which is related to the inertia of the fluid and happens at very short times. For the case of deionized water, the ions would be mainly H^+ and OH^- , whose diffusion coefficients are of the order of $5-10 \cdot 10^{-9} \text{ m}^2/\text{s}$, returning a $\tau_\alpha \simeq 1 \text{ ms}$ for the $5 \mu\text{m}$ particles used in the experiment. The other timescales are shorter. I guess the problem is still more complicated, and the relaxation in the trap will be more involved. There are a few publications on the electrophoresis of trapped particles that could be considered for a better understanding of the problem:

<https://doi.org/10.1016/j.jcis.2009.05.017>

<https://doi.org/10.1063/1.2884147>

<https://doi.org/10.1002/elps.201300214>

<https://doi.org/10.1063/1.4882419>

<https://doi.org/10.1103/PhysRevE.90.032116>

I understand that this is too much information to be included in the manuscript, but I definitely ask the Authors to at least indicate that the problem is more complex than just the single timescale they wrongly identify.

Response: We thank the reviewers for their comments and for recommending the paper for publication subject to changes. We apologize for the confusion with the mechanisms responsible for the electrophoretic noise. We thank the reviewer for the detailed explanation of the complex nature of the noise involved in our problem. We have now rewritten the supplementary information to describe the processes as detailed by the reviewer and included the appropriate references.

Changes made in lines 52-54 of the main manuscript and lines 16-30, 42-56 of the supplementary information. We have included references 23,24 in the main text and 1, 4-10 in the supplementary information.

Reviewer #3 (Remarks to the Author):

As indicated in my previous reports in Nature Physics, I find the results, methodology, and impact of the paper suitable for publication in Nature Communications, and even more after the new modifications in the manuscript.

The authors have successfully tackled most of my previous queries and improve the clarity and interpretation of their excellent experimental results. Now it is more clear the theoretical background and why the results are interesting for a broad class of physicists.

The manuscript has gained a lot especially in what refers to quantification of non-Markovianity, which is key, and is now provided in terms of a recently-introduced measure. I just have one suggestion for the authors, namely include Fig S2C in the Main Text, either as a panel or as an inset, or even the three panels in S2. I find the result in S2C excellent, and a perfect fingerprint of non-Markovianity in their experimental setup, which is a central assumption of the authors' claim, and in my opinion deserves to be revealed somewhere in the Main Text.

Once this modification is implemented I am ready to recommend the article for publication in Nature Communications.

Response: We thank the reviewer for the comments and recommending the article for publication subject to modification. We apologize for not including the recently-introduced measure of non-Markovianity in the main text. Following the reviewer's comments, we have now moved Fig. S2 into Fig. 2 of the main text. We hope that the revised manuscript highlights the non-Markovian nature of the noise appropriately.

Fig. 2 added and changes made in lines 59-69 and 72-83 of the main text.